

SciPost Phys. Lect. Notes 57 (2022)

# Quantum to classical mapping
# of the two-dimensional toric code in an external field

**Sydney R. Timmerman[1], Zvonimir Z. Bandić[2] and Roger G. Melko[1,3]⋆**

**1** Perimeter Institute for Theoretical Physics, Waterloo, Ontario N2L 2Y5, Canada
**2** Western Digital Research, San Jose, CA 95119, USA
**3** Department of Physics and Astronomy, University of Waterloo, Ontario N2L 3G1, Canada

⋆ rmelko@perimeterinstitute.ca

## Abstract

Kitaev's toric code Hamiltonian in dimension $D = 2$ has been extensively studied for its topological properties, including its quantum error correction capabilities. While the Hamiltonian is quantum, it lies within the class of models that admits a $D+1$-dimensional classical representation. In these notes, we provide details of a Suzuki-Trotter expansion of the partition function of the toric code Hamiltonian in the presence of an external magnetic field. By coupling additional degrees of freedom in the form of a matter field that can subsequently be gauged away, we explicitly derive a classical Hamiltonian on a cubic lattice which takes the form of a non-isotropic $3D$ Ising gauge theory.

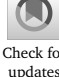
## Contents

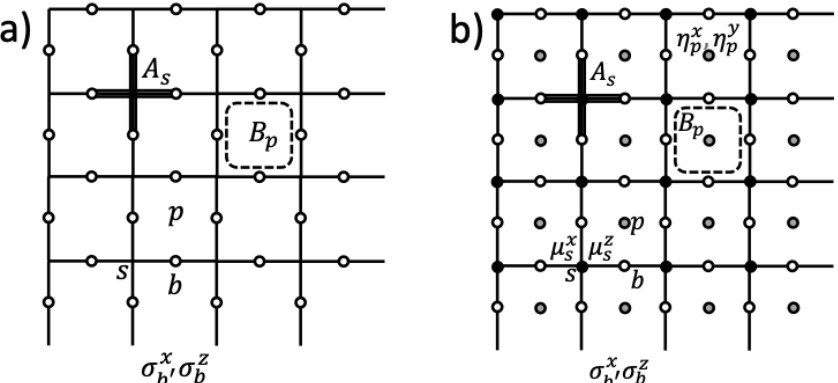

Figure 1: A 2$D$ square lattice with periodic boundary conditions. a) Labels for lattice sites $s$, bonds $b$ and plaquettes $p$ are shown, as well as spins involved in site operators $\hat{A}_s$ and plaquette operators $\hat{B}_p$. b) Redundant 1/2 spin degrees of freedom $\hat{\vec{\mu}}_s$ and $\hat{\vec{\eta}}_p$ introduced at the sites $s$ and center of plaquettes face $p$ are shown. We couple Kitaev's toric code Hamiltonian in an external magnetic field to the spins $\hat{\vec{\eta}}_p$.

## 1 Introduction

Kitaev's famous Hamiltonian, also referred to as the *toric code*, has captured the attention of a broad community and defined a once-in-a-generation paradigm surrounding the physics of deconfinement, topological order and quantum error correction [1]. The toric code Hamiltonian is an important tool since it contains the simplest topologically-ordered phase – the deconfined $\mathbb{Z}_2$ quantum spin liquid – with gapped anyonic excitations that play an important role in proposals for topological quantum computing [2], and can be condensed to quantum critical points that display universal physics. Importantly, the toric code can be modified with a number of additional Hamiltonian terms which greatly enrich its physics while remaining simple to analyze in various limits. While the toric code is explicitly quantum, its partition function in two spatial dimensions admits a mapping to a three-dimensional (3$D$) classical partition function that can be further analyzed with analytical or numerical techniques [3,4]. In these notes, we provide a detailed derivation of this mapping.

Kitaev defined the Hamiltonian of the toric code as,

$$\hat{H}_{TC} = -\sum_s \hat{A}_s - \sum_p \hat{B}_p, \tag{1}$$

where the spin-1/2 degrees of freedom $\hat{\vec{\sigma}}$ are located on the bonds of the two-dimensional (2$D$) square lattice placed on the torus. The operators $\hat{A}_s$ and $\hat{B}_p$ are given by $\hat{A}_s = \prod_{j\in s} \hat{\sigma}_j^x$ and $\hat{B}_p = \prod_{j\in p} \hat{\sigma}_j^z$, where $s$ represents the site of the lattice and $p$ represent plaquettes on the lattice (see Fig.1a).

The ground state solution of the toric code is easy to obtain, since the operators $\hat{A}_s$ and $\hat{B}_p$ commute. The Hamiltonian has eigenvalues $\hat{A}_s = 1$ and $\hat{B}_p = 1$ for all $s$ and $p$. It is four-fold degenerate on a 2$D$ torus, with gapped elementary excitations. These excitations are represented by $\hat{A}_s = -1$ and $\hat{B}_p = -1$, which can also be viewed as a $\mathbb{Z}_2$ electric charge on site $s$ and $\mathbb{Z}_2$ magnetic charge (vortex) on plaquette $p$.

In previous studies it has been shown that the topological ground state of the toric code is robust against longitudinal magnetic field perturbations of the form $-h\sum_b \hat{\sigma}_b^z$ [5] and more generally both longitudinal and transverse fields $-h_x\sum_b \hat{\sigma}_b^x - h_z\sum_b \hat{\sigma}_b^z$ [6]. Considering electric and magnetic charge conservations laws [1], Kitaev introduced additional spin degrees of

freedom (or matter fields) $\hat{\vec{\mu}}_s$ and $\hat{\vec{\eta}}_p$ for each vertex $s$ and plaquette $p$. The additional spins contribute to a unique quantum state $|\zeta\rangle$ such that $\hat{\mu}_s^x |\zeta\rangle = |\zeta\rangle$ and $\hat{\eta}_p^z |\zeta\rangle = |\zeta\rangle$. We should observe that introduction of the new spin degrees of freedom does not change the form of the Hamiltonian $\hat{H}_{TC}$, as it does not contain any terms coupled with $\hat{\vec{\mu}}_s$ and $\hat{\vec{\eta}}_p$. This leads to an embedding of the Hilbert space $\mathcal{N}$ of spins $|\psi\rangle = |\sigma_b^z\rangle \otimes |\sigma_b^x\rangle$ in $\hat{H}_{TC}$ into a larger Hilbert space $\mathcal{T}$ of all the spins $|\psi\rangle \mapsto |\psi\rangle \otimes |\zeta\rangle$. The physical states $\psi \in \mathcal{N}$ also satisfy $\hat{\mu}_s^x |\psi\rangle = |\psi\rangle$ and $\hat{\eta}_p^z |\psi\rangle = |\psi\rangle$, as $\hat{H}_{TC}$ does not depend on any terms that use the additional degrees of freedom $\hat{\vec{\mu}}_s$ and $\hat{\vec{\eta}}_p$.

A growing number of studies, particularly those that wish to use numerical techniques like Monte Carlo [6–8], exploit a quantum-to-classical mapping of the $2D$ toric code. A straightforward way to derive this mapping is to use the Suzuki-Trotter expansion [9] of the partition function to derive a classical partition function and the corresponding $3D$ classical Hamiltonian. The Suzuki-Trotter expansion can be done in multiple different ways. The key to the procedure being analytically tractable is to pick the basis of the expansion in such a way that the Hamiltonian can be written as a sum of diagonal and off-diagonal components, and that the portion of the partition function corresponding to the off-diagonal component of the Hamiltonian can be analytically computed. One example of such analytically tractable off-diagonal Hamiltonian is a linear combination of spin degrees of freedom without any higher order terms. This is exactly what motivates our following procedure in which we will transform the perturbed Hamiltonian: $\hat{H} = -\sum_s \hat{A}_s - \sum_p \hat{B}_p - h_x \sum_b \hat{\sigma}_b^x - h_z \sum_b \hat{\sigma}_b^z$ into a form containing an off-diagonal component that is linear in all terms. Of course, this does not preclude that other analytical methods are possible. In this paper, we follow the procedure of coupling the toric code Hamiltonian to the additional spins discussed above, and then gauging the resulting Hamiltonian in order to produce a form that can be integrated in the Suzuki-Trotter expansion. To derive a $3D$ classical Hamiltonian that produces the same partition function as its quantum equivalent, we will employ the following steps. First, we pick an array of redundant spins (i.e. not explicitly present in the $\hat{H}_{TC}$) with the same symmetry as the original array of spins. In our case, these are spins $\hat{\vec{\eta}}_p$ in the center of each plaquette as discussed above. We then extend the Hilbert space to include redundant spins, and observe that the energy spectrum is not affected, as $\hat{H}_{TC}$ does not contain $\hat{\vec{\eta}}_p$. After this step, we are allowed to perform unitary operations in the extended Hilbert space - these will also not affect the energy spectrum that we are trying to understand, but will couple redundant spins $\hat{\vec{\eta}}_p$ with spins $\hat{\vec{\sigma}}_b$. At this point, we will seek specific choices of unitary operators $\mathcal{U}$ whose symmetry operator $\hat{Q}_p$ generates the $\mathbb{Z}_2$ gauge transformation. This specific choice will give us the gauge freedom to transform the Kitaev toric code Hamiltonian in an external magnetic field into a form that can be explicitly integrated via a Suzuki-Trotter expansion.

As mentioned, the process of embedding in a larger Hilbert space does not change the energy spectrum (as spins $\hat{\vec{\mu}}_s$ and $\hat{\vec{\eta}}_p$ are not explicitly present in $\hat{H}_{TC}$). However, the newly gained gauge freedom does allow us to apply unitary transformations $\mathcal{U}$ on top of the extended Hilbert space $\mathcal{T}$, that would couple the toric code spins $\hat{\sigma}_b^z$ and $\hat{\sigma}_b^x$ with the additional spin degrees of freedom $\hat{\vec{\mu}}_s$ and $\hat{\vec{\eta}}_p$. One possible coupling with $\hat{\vec{\mu}}_s$ is described by Tupitsyn $et\ al.$ in Ref. [6]. In this study, we focus on coupling with plaquette-centered spin degrees of freedom $\hat{\vec{\eta}}_p$, by considering a unitary transformation $\mathcal{U}$ that performs the map,

$$\hat{\sigma}_b^{x\prime} \mapsto \hat{\eta}_{p_1}^x \hat{\sigma}_b^x \hat{\eta}_{p_2}^x. \tag{2}$$

With this mapping, the physical subspace becomes $\mathcal{N}' = \mathcal{U}\mathcal{N}$, and vectors belonging to $\mathcal{N}'$ are invariant under the symmetry operator $\hat{Q}_p = \hat{U}\hat{\sigma}_p^z\hat{U}^\dagger = \hat{\eta}_p^z\hat{B}_p$. Since the transformed Hamiltonian $\hat{H}'_{TC} = \hat{U}\hat{H}_{TC}\hat{U}^\dagger$ commutes with the symmetry operator $\hat{Q}_p$, and since $\hat{Q}_p^2 = 1$ and $\hat{Q}_p$ operators commute amongst themselves $[\hat{Q}_{p_1}, \hat{Q}_{p_2}] = 0$, the operator $\hat{Q}_p$ generates a

$\mathbb{Z}_2$ gauge transformation; we call it a generator $\hat{Q}_p$ or magnetic gauge [10]. The states $\psi' \in \mathcal{N}'$ are thus eigenstates of both $\hat{H}'_{TC}$ and of all the generators $\hat{Q}_p$. We can then define the new Hilbert space of gauge invariant states by making a choice of $\hat{Q}_p = 1$ or $\hat{Q}_p |\psi'\rangle = |\psi'\rangle$ [10]. This is a very convenient choice, because it will help us simplify the terms that perturb $\hat{H}_{TC}$, i.e. in the $\hat{Q}_p = \mathbb{1}$ Hilbert space, the terms proportional to $\hat{B}_p$ can be replaced with terms proportional to $\hat{\eta}^z_p$. In this new Hilbert space $\hat{Q}_p = \mathbb{1}$ represents a form of the familiar Gauss-law condition, but instead of $\hat{B}_p = \mathbb{1}$ we have dynamical source in the form of $\hat{\eta}^z_p$.

Therefore, we proceed to analyze the Kitaev toric code with external magnetic fields:

$$\hat{H} = -J_x \sum_s \hat{A}_s - J_z \sum_p \hat{B}_p - h_x \sum_b \hat{\sigma}^x_b - h_z \sum_b \hat{\sigma}^z_b, \tag{3}$$

by considering Hilbert space $\hat{Q}_p = \mathbb{1}$ (or working in a magnetic gauge $\hat{\eta}^z_p \hat{B}_p = \mathbb{1}$). The Hamiltonian of Eq. (3) together with Eq. (2) and in the $\hat{Q}_p = \mathbb{1}$ gauge becomes:

$$\hat{H}' = -h_x \sum_{p,q} \hat{\eta}^x_p \hat{\sigma}^x_{pq} \hat{\eta}^x_q - J_x \sum_s \hat{A}_s - J_z \sum_p \hat{\eta}^z_p - h_z \sum_b \hat{\sigma}^z_b. \tag{4}$$

We will now analyze this Hamiltonian using an explicit quantum-to-classical mapping.

## 2 Suzuki-Trotter decomposition of the toric code in the $\hat{Q}_p = \mathbb{1}$ gauge

We will now focus on computing the quantum statistical partition function $\mathcal{Z} = Tr\{e^{-\beta \hat{H}'}\}$ of Hamiltonian Eq. (4), under gauge constraint $\hat{\eta}^z_p \hat{B}_p = \mathbb{1}$. Notice that, in this gauge, the portion of the Hamiltonian diagonal in the $z$-basis $\hat{H}_z = -J_z \sum_p \hat{\eta}^z_p - h_z \sum_b \hat{\sigma}^z_b$ no longer has any terms dependent on products of spins , while the portion $\hat{H}_x = -h_x \sum_{p,q} \hat{\eta}^x_p \hat{\sigma}^x_{pq} \hat{\eta}^x_q - J_x \sum_s \hat{A}_s$ , diagonal in $x$-basis, does have terms dependent on products of spins . To take full advantage of this simplification, we will perform the Suzuki-Trotter decomposition in the $x$-basis $\{\sigma^x_b\} \otimes \{\eta^x_p\}$, so that $\hat{H}_z$ is the off-diagonal part of the Hamiltonian. That way, the term $\hat{H}_x$ is diagonal, and off-diagonal $\hat{H}_z$ does not have terms dependent on products of spins, making it tractable for computation of partition function.

Following usual prescription for the Suzuki-Trotter expansion that

$$e^{-\beta \hat{H}'} = \lim_{M \to \infty} (e^{-\Delta\tau \hat{H}'})^M, \tag{5}$$

where $\Delta\tau = \frac{\beta}{M}$, we have

$$\mathcal{Z} = \sum_{\{\sigma^x_b, \eta^x_p\}} \langle \{\sigma^x_b\} \otimes \{\eta^x_p\}| \underbrace{\left(e^{-\varepsilon \hat{H}'}\right)\left(e^{-\varepsilon \hat{H}'}\right)...\left(e^{-\varepsilon \hat{H}'}\right)}_{M \text{ factors}}$$
$$\times \prod_q \delta(\mathbb{1} - \hat{\eta}_q \hat{B}_q) |\{\sigma^x_b\} \otimes \{\eta^x_p\}\rangle, \tag{6}$$

where the operators $\delta(\mathbb{1} - \hat{\eta}_q \hat{B}_q)$ are necessary to enforce the gauge condition. Inserting the identity $\mathbb{1} = \sum_{\{\sigma^x_b, \eta^x_p\}} |\{\sigma^x_b\} \otimes \{\eta^x_p\}\rangle \langle \{\sigma^x_b\} \otimes \{\eta^x_p\}|$ between each of the $M$ factors, this

becomes,

$$
\mathcal{Z} = \left[\prod_{k=0}^{M-1} \sum_{\{\sigma_b^x(k\varepsilon),\eta_p^x(k\varepsilon)\}}\right]
$$
$$
\times \langle\{\sigma_b^x\}\otimes\{\eta_p^x\}(0)|\left(e^{-\varepsilon\hat{H}'}\right)\prod_q \delta(\mathbb{1}-\hat{\eta}_q\hat{B}_q)|\{\sigma_b^x\}\otimes\{\eta_p^x\}((M-1)\varepsilon)\rangle
$$
$$
\times \langle\{\sigma_b^x\}\otimes\{\eta_p^x\}((M-1)\varepsilon)|\left(e^{-\varepsilon\hat{H}'}\right)\prod_q \delta(\mathbb{1}-\hat{\eta}_q\hat{B}_q)...
$$
$$
\times \langle\{\sigma_b^x\}\otimes\{\eta_p^x\}(\varepsilon)|\left(e^{-\varepsilon\hat{H}'}\right)\prod_q \delta(\mathbb{1}-\hat{\eta}_q\hat{B}_q)|\{\sigma_b^x\}\otimes\{\eta_p^x\}(0)\rangle \,,
\tag{7}
$$

where we distinguish the $M-1$ additional bases by labeling them with a parameter $\tau$. Because $e^{-\varepsilon\hat{H}'}$ is the Euclidean time-evolution operator for time interval $\varepsilon$, by augmenting $\tau$ by $\varepsilon$ after each factor, we can interpret $\tau$ as a Euclidean time coordinate, labeling bases at different imaginary times.

In more compact notation,

$$
\mathcal{Z} = \left[\prod_{k=0}^{M-1} \sum_{\{\sigma_b^x(k\varepsilon),\eta_p^x(k\varepsilon)\}}\right]\prod_{l=0}^{M-1} \langle\{\sigma_b^x\}\otimes\{\eta_p^x\}((l+1)\varepsilon)|\left(e^{-\varepsilon\hat{H}'}\right)
$$
$$
\times \prod_q \delta(\mathbb{1}-\hat{\eta}_q\hat{B}_q)|\{\sigma_b^x\}\otimes\{\eta_p^x\}(l\varepsilon)\rangle|_{tr}\,,
\tag{8}
$$

where we use $|_{tr}$ to denote the condition that $\langle\{\sigma_b^x\}\otimes\{\eta_p^x\}(M\varepsilon)| = \langle\{\sigma_b^x\}\otimes\{\eta_p^x\}(0)|$, originating from the trace. Since $\hat{H}' = \hat{H}_x + \hat{H}_z$ we can apply the Baker–Campbell–Hausdorff (BCH) formula, which simplifies the term $e^{-\varepsilon\hat{H}'} \simeq e^{-\varepsilon\hat{H}_x}e^{-\varepsilon\hat{H}_z}$, with the leading correction term proportional to $-\frac{1}{2}\varepsilon^2[\hat{H}_x,\hat{H}_z]$. So the rest of the calculation will be correct up to the order of $\varepsilon^2$, which is acceptable for $M \gg 1$ limit (see Eq. 5). Expanding out the Hamiltonian, this is

$$
\mathcal{Z} = \left[\prod_{k=0}^{M-1} \sum_{\{\sigma_b^x(k\varepsilon),\eta_p^x(k\varepsilon)\}}\right]Z_{diag}\prod_{l=0}^{M-1} \langle\{\sigma_b^x\}\otimes\{\eta_p^x\}((l+1)\varepsilon)|\left(e^{\varepsilon J_z \sum_p \hat{\eta}_p^z}e^{\varepsilon h_z \sum_b \hat{\sigma}_b^z}\right)
$$
$$
\times \prod_q \delta(\mathbb{1}-\hat{\eta}_q\hat{B}_q)|\{\sigma_b^x\}\otimes\{\eta_p^x\}(l\varepsilon)\rangle|_{tr}\,.
\tag{9}
$$

Here, the term $Z_{diag}$ refers to contribution to the partition function of the diagonal part of the Hamiltonian, $\hat{H}_x = -h_x \sum_{p,q} \hat{\eta}_p^x \hat{\sigma}_{pq}^x \hat{\eta}_q^x - J_x \sum_s \hat{A}_s$. Since $\hat{H}_x$ is diagonal in the $\{\sigma_b^x\}\otimes\{\eta_p^x\}$ basis, its sumation is trivial, and $Z_{diag}$ is factored out in the partition function. The interesting terms that we need to handle in Eq. (9) are the Kronecker delta symbols $\delta(\mathbb{1}-\hat{\eta}_q\hat{B}_q)$, which ensure the correct projection onto the Hilbert space defined by the $\hat{Q}_p = \mathbb{1}$ gauge, and exponential terms $e^{\varepsilon J_z \sum_p \hat{\eta}_p^z}$ and $e^{\varepsilon h_z \sum_b \hat{\sigma}_b^z}$ that appear challenging for summation.

## 3 The classical Ising gauge theory in $3D$

To evaluate the Suzuki-Trotter decomposition of the quantum statistical partition function in Eq. (9), we will take advantage of some identities relating Pauli spin operators, which are shown and proven in the appendices. We can use Eq. (27) to rewrite $\delta(\mathbb{1}-\hat{\eta}_q\hat{B}_q)$, the term which ensures correct projection into the Hilbert space defined by the $\hat{Q}_p = \mathbb{1}$ gauge, for each bond and each plaquette. Eq. (27) introduces sums over new classical dummy "spins"

associated with each plaquette, $s_p^x$, which effectively expands our Hilbert space again from $\{\sigma_b^x\} \otimes \{\eta_p^x\} \mapsto \{\sigma_b^x\} \otimes \{\eta_p^x\} \otimes \{s_p^x\}$. Additionally, we will insert the form of the identity given in Eq. (28) twice, once for the $\hat{\vec{\eta}}$ degrees of freedom, and once for the $\hat{\vec{\sigma}}$. This gives,

$$
\begin{aligned}
\mathcal{Z} = \left[ \prod_{k=0}^{M-1} \sum_{\{\sigma_{pq}^x \otimes \eta_p^x\}(k\varepsilon)} \right] \mathcal{Z}_{diag} \prod_{l=0}^{M-1} &\langle \{\sigma_{pq}^x \otimes \eta_p^x\}((l+1)\varepsilon)| \\
&\times e^{i\frac{\pi}{2} \sum_{pq}(1-\hat{\sigma}_{pq}^z)\left[\left(1-\sigma_{pq}^x(l\varepsilon)\right)+\left(1-\sigma_{pq}^x((l+1)\varepsilon)\right)\right]} e^{\varepsilon h_z \sum_{pq} \hat{\sigma}_{pq}^z} \\
&\times e^{i\frac{\pi}{2} \sum_p(1-\hat{\eta}_p^z)\left[\left(1-\eta_p^x(l\varepsilon)\right)+\left(1-\eta_p^x((l+1)\varepsilon)\right)\right]} e^{\varepsilon J_z \sum_p \hat{\eta}_p^z} \\
&\times \frac{1}{2} \prod_q \sum_{s_q^x(l\varepsilon)=\pm 1} e^{i\pi \frac{1-s_q^x(l\varepsilon)}{2}\left[\frac{1-\hat{\eta}_q^z}{2}+\sum_{pq\in q}\frac{1-\hat{\sigma}_{pq}^z}{2}\right]} \\
&\times |\{\sigma_{pq}^x \otimes \eta_p^x\}(l\varepsilon)\rangle \Big|_{tr} ,
\end{aligned}
\tag{10}
$$

where we have parameterized the new classical degrees of freedom $s_q^x$ with the Euclidean time as we have inserted the identity Eq. (27) once at each time.

Folding the sums over $s_q^x(l\varepsilon)$ in with the other sums over spin configurations, and pulling the sums over bonds and plaquettes out of the exponentials, we have

$$
\begin{aligned}
\mathcal{Z} = \left[ \prod_{k=0}^{M-1} \sum_{\{\sigma_{pq}^x \otimes \eta_p^x, s_q^x\}(k\varepsilon)} \right] \mathcal{Z}_{diag} \prod_{l=0}^{M-1} &\langle \{\sigma_{pq}^x \otimes \eta_p^x\}((l+1)\varepsilon)| \\
&\times \prod_{pq} \left( e^{i\frac{\pi}{2}(1-\hat{\sigma}_{pq}^z)\left[\left(1-\sigma_{pq}^x(l\varepsilon)\right)+\left(1-\sigma_{pq}^x((l+1)\varepsilon)\right)\right]} e^{\varepsilon h_z \hat{\sigma}_{pq}^z} \right) \\
&\times \prod_p \left( e^{i\frac{\pi}{2}(1-\hat{\eta}_p^z)\left[\left(1-\eta_p^x(l\varepsilon)\right)+\left(1-\eta_p^x((l+1)\varepsilon)\right)\right]} e^{\varepsilon J_z \hat{\eta}_p^z} \right) \\
&\times \frac{1}{2} \prod_q e^{i\pi \frac{1-s_q^x(l\varepsilon)}{2}\left[\frac{1-\hat{\eta}_q^z}{2}+\sum_{pq\in q}\frac{1-\hat{\sigma}_{pq}^z}{2}\right]} |\{\sigma_{pq}^x \otimes \eta_p^x\}(l\varepsilon)\rangle \Big|_{tr} .
\end{aligned}
\tag{11}
$$

We can now regroup terms so that we only have two products inside of the matrix element, one for each bond and one for each plaquette, in anticipation of the fact that our final classical Hamiltonian will be comprised of sums over bonds and sums over plaquettes. This gives

$$
\begin{aligned}
\mathcal{Z} = \left[ \prod_{k=0}^{M-1} \sum_{\{\sigma_{pq}^x \otimes \eta_p^x, s_q^x\}(k\varepsilon)} \right] \mathcal{Z}_{diag} \prod_{l=0}^{M-1} &\langle \{\sigma^{pq} \otimes \eta_p^x\}((l+1)\varepsilon)| \\
&\times \prod_{pq} \left( e^{i\frac{\pi}{2}(1-\sigma_{pq}^z)\left[\left(1-\sigma_{pq}^x(l\varepsilon)\right)+\left(1-\sigma_{pq}^x((l+1)\varepsilon)\right)+\frac{1-s_p^x(l\varepsilon)}{2}+\frac{1-s_q^x(l\varepsilon)}{2}\right]} e^{\varepsilon h_z \sigma_{pq}^z} \right) \\
&\times \prod_p \left( e^{i\frac{\pi}{2}(1-\eta_p^z)\left[\left(1-\eta_p^x(l\varepsilon)\right)+\left(1-\eta_p^x((l+1)\varepsilon)\right)+\frac{1-s_p^x(l\varepsilon)}{2}\right]} e^{\varepsilon J_z \eta_p^z} \right) \\
&\times |\{\sigma_{pq}^x \otimes \eta_p^x\}(l\varepsilon)\rangle \Big|_{tr} ,
\end{aligned}
\tag{12}
$$

where $s^x$ appears twice in the product over bonds because each bond is between two plaquettes.

We have now lost the complex projection terms and put the partiton function into a form simple enough to evaluate directly by using Eq. (26) and making an ansatz. First, notice that

the factor $\left[\left(1-\eta_p^x(l\varepsilon)\right)+\left(1-\eta_p^x((l+1)\varepsilon)\right)+\frac{1-s_p^x(l\varepsilon)}{2}\right]$ can only be odd if $s_p^x(l\varepsilon)=-1$, as the $x$-basis eigenvalues, $\eta_p^x(l\varepsilon),\eta_p^x((l+1)\varepsilon),s_p^x(l\varepsilon)=\pm1$. This implies that, as in the derivation of spin identity Eq. (26), the term inside Eq. (12) can be replaced with

$$e^{i\frac{\pi}{2}(1-\hat{\eta}_p^z)\left[\left(1-\eta_p^x(l\varepsilon)\right)+\left(1-\eta_p^x((l+1)\varepsilon)\right)+\frac{1-s_p^x(l\varepsilon)}{2}\right]}=\begin{cases}\mathbb{1}&\text{when }s_p^x(l\varepsilon)=1\,,\\\hat{\eta}_p^z&\text{when }s_p^x(l\varepsilon)=-1\,.\end{cases}\tag{13}$$

Returning to the full plaquette term in Eq. (12), we can decompose the exponential $e^{\varepsilon J_z\eta_p^z}$ in terms of trigonometric functions to get

$$\begin{aligned}e^{\varepsilon J_z\hat{\eta}_p^z}&=\cosh(\varepsilon J_z\hat{\eta}_p^z)+\sinh(\varepsilon J_z\hat{\eta}_p^z)\\&=\cosh(\varepsilon J_z)\mathbb{1}+\sinh(\varepsilon J_z)\hat{\eta}_p^z\,,\end{aligned}\tag{14}$$

where in the second equality we have used the fact that the eigenvalues of $\hat{\eta}_p^z$ are $\pm1$ and that $\cosh(x)$ is an odd function while $\sinh(x)$ is even. Putting this together with Eq. (13), we see

$$e^{i\frac{\pi}{2}(1-\hat{\eta}_p^z)\left[\left(1-\eta_p^x(l\varepsilon)\right)+\left(1-\eta_p^x((l+1)\varepsilon)\right)+\frac{1-s_p^x(l\varepsilon)}{2}\right]}e^{\varepsilon J_z\hat{\eta}_p^z}$$

$$=\begin{cases}\cosh(\varepsilon J_z)\mathbb{1}+\sinh(\varepsilon J_z)\hat{\eta}_p^z&\text{when }s_p^x(l\varepsilon)=1\,,\\\cosh(\varepsilon J_z)\hat{\eta}_p^z+\sinh(\varepsilon J_z)\mathbb{1}&\text{when }s_p^x(l\varepsilon)=-1\,.\end{cases}$$

Evaluating this within the matrix element gives

$$\langle\eta_p^x((l+1)\varepsilon)|\left(e^{i\frac{\pi}{2}(1-\hat{\eta}_p^z)\left[\left(1-\eta_p^x(l\varepsilon)\right)+\left(1-\eta_p^x((l+1)\varepsilon)\right)+\frac{1-s_p^x(l\varepsilon)}{2}\right]}e^{\varepsilon J_z\hat{\eta}_p^z}\right)|\eta_p^x(l\varepsilon)\rangle$$

$$=\begin{cases}\cosh(\varepsilon J_z)&\text{when }s_p^x(l\varepsilon)\eta_p^x((l+1)\varepsilon)\eta_p^x(l\varepsilon)=1\\\sinh(\varepsilon J_z)&\text{when }s_p^x(l\varepsilon)\eta_p^x((l+1)\varepsilon)\eta_p^x(l\varepsilon)=-1\end{cases}\tag{15}$$

$$\equiv A_2e^{k_2s_p^x(l\varepsilon)\eta_p^x((l+1)\varepsilon)\eta_p^x(l\varepsilon)}\,.$$

as, when $\eta_p^x((l+1)\varepsilon)$ and $\eta_p^x(l\varepsilon)$ are the same, that is $\eta_p^x((l+1)\varepsilon)\eta_p^x(l\varepsilon)=1$, the term proportional to $\mathbb{1}$ will survive. On the other hand, when $\eta_p^x((l+1)\varepsilon)\eta_p^x(l\varepsilon)=-1$, the term proportional to $\hat{\eta}_p^z$ will flip $\eta_p^x(l\varepsilon)$ and contribute. Put together with the dependence on $s_p^x(l\varepsilon)$, we find that this part of the matrix element, and therefore this portion of the 3$D$ classical Hamiltonian, depends only on the product of all three eigenvalues. We can solve it by making the ansatz shown in the second equality, finding

$$A_2=\left(\sinh(\varepsilon J_z)\cosh(\varepsilon J_z)\right)^{1/2}\tag{16}$$

$$k_2=-\frac{1}{2}\ln\tanh(\varepsilon J_z)\,.\tag{17}$$

This is the step when all of the spin variables inside the quantum partition function move into the exponent, allowing us to analytically identify the emerging 3$D$ classical Hamiltonian; we started with $s_p$ as a dummy spin variable, but through this process it has been promoted to an actual classical spin.

Returning to the matrix element in Eq. (12), all that remains to be computed are the factors associated to each bond $pq$, which depend on the $\hat{\sigma}$ spins. We can evaluate the matrix element on these factors by following exactly the same procedure as above, finding

$$e^{i\frac{\pi}{2}(1-\hat{\sigma}_{pq}^z)\left[\left(1-\sigma_{pq}^x(l\varepsilon)\right)+\left(1-\sigma_{pq}^x((l+1)\varepsilon)\right)+\frac{1-s_p^x(l\varepsilon)}{2}+\frac{1-s_q^x(l\varepsilon)}{2}\right]}e^{\varepsilon h_z\hat{\sigma}_{pq}^z}$$

$$=\begin{cases}\cosh(\varepsilon h_z)\mathbb{1}+\sinh(\varepsilon h_z)\hat{\sigma}_{pq}^z&\text{when }s_p^x(l\varepsilon)s_q^x(l\varepsilon)=1\,,\\\cosh(\varepsilon h_z)\hat{\sigma}_{pq}^z+\sinh(\varepsilon h_z)\mathbb{1}&\text{when }s_p^x(l\varepsilon)s_q^x(l\varepsilon)=-1\,.\end{cases}$$

The only small difference from the previous case with the $\hat{\vec{\eta}}$ spins arises because there are two $s^x$ terms in the bracketed factor on the first line—the bracketed factor can only be odd if just one of $s_p^x(l\varepsilon)$ and $s_q^x(l\varepsilon)$ is $-1$, meaning that the cases are distinguished by the *product* $s_p^x(l\varepsilon)s_q^x(l\varepsilon)$. Evaluating this within the matrix element, we have,

$$
\langle \sigma_{pq}^x((l+1)\varepsilon)|
$$
$$
\times \left( e^{i\frac{\pi}{2}(1-\hat{\sigma}_{pq}^z)\left[\left(1-\sigma_{pq}^x(l\varepsilon)\right)+\left(1-\sigma_{pq}^x((l+1)\varepsilon)\right)+\frac{1-s_p^x(l\varepsilon)}{2}+\frac{1-s_q^x(l\varepsilon)}{2}\right]}e^{\varepsilon h_z \hat{\sigma}_{pq}^z} \right) |\sigma_{pq}^x(l\varepsilon)\rangle
$$
$$
= \begin{cases} \cosh(\varepsilon h_z) & \text{when } s_p^x(l\varepsilon)s_q^x(l\varepsilon)\eta_p^x((l+1)\varepsilon)\eta_p^x(l\varepsilon)=1 \\ \sinh(\varepsilon h_z) & \text{when } s_p^x(l\varepsilon)s_q^x(l\varepsilon)\sigma_{pq}^x((l+1)\varepsilon)\sigma_{pq}^x(l\varepsilon)=-1 \end{cases}
$$
$$
\equiv A_2' e^{k_2' s_p^x(l\varepsilon)s_q^x(l\varepsilon)\sigma_{pq}^x((l+1)\varepsilon)\sigma_{pq}^x(l\varepsilon)},
$$

where we have found that the classical Hamiltonian now depends on the product of all *four* eigenvalues, and made the corresponding ansatz in the second equality. It is solved by

$$
A_2' = \left( \sinh(\varepsilon h_z)\cosh(\varepsilon h_z) \right)^{1/2} \tag{18}
$$
$$
k_2' = -\frac{1}{2}\ln\tanh(\varepsilon h_z). \tag{19}
$$

Now that we have fully evaluated the matrix element, we can substitute our solution back into Eq. (12) to find the now fully-classical partition function,

$$
\mathcal{Z} = \left[ \prod_{k=0}^{M-1} \sum_{\{\sigma_{pq}^x,\eta_p^x,s_q^x\}(k\varepsilon)} \right] \mathcal{Z}_{diag} \prod_{l=0}^{M-1} \prod_p A_s e^{k_2 s_p^x(l\varepsilon)\eta_p^x((l+1)\varepsilon)\eta_p^x(l\varepsilon)}
$$
$$
\times \prod_{pq} A_2' e^{k_2' s_p^x(l\varepsilon)s_q^x(l\varepsilon)\sigma_{pq}^x((l+1)\varepsilon)\sigma_{pq}^x(l\varepsilon)} \Bigg|_{tr}. \tag{20}
$$

Expanding out the contribution to $\mathcal{Z}$ from the previously-diagonal $\hat{H}_x$ and massaging to read off the classical Hamiltonian, we have

$$
\mathcal{Z} = A_2^N A_2'^N \left[ \prod_{k=0}^{M-1} \sum_{\{\sigma_{pq}^x,\eta_p^x,s_q^x\}(k\varepsilon)} \right] e^{\sum_l \sum_{pq} k_2' s_p^x(l\varepsilon)s_q^x(l\varepsilon)\sigma_{pq}^x((l+1)\varepsilon)\sigma_{pq}^x(l\varepsilon)}
$$
$$
\times e^{\sum_l \sum_p k_2 s_p^x(l\varepsilon)\eta_p^x((l+1)\varepsilon)\eta_p^x(l\varepsilon)}
$$
$$
\times e^{\sum_l (\varepsilon J_x \sum_s \prod_{p\in s}\sigma_p^x(l\varepsilon)+\varepsilon h_x \sum_{pq}\eta_p^x(l\varepsilon)\sigma_{pq}^x(l\varepsilon)\eta_q^x)(l\varepsilon))} \Bigg|_{tr}, \tag{21}
$$

where we have moved the products over bonds and plaquettes, as well as the product over the discrete time parameter $l$, into the exponent, where they become spatial and temporal sums.

The 3$D$ classical Hamiltonian is then

$$
\beta' H_c = -\sum_{l=0}^{M-1} \left( \varepsilon J_x \sum_s \prod_{p\in s}\sigma_p^x(l\varepsilon) + k_2' \sum_{pq} s_p^x(l\varepsilon)s_q^x(l\varepsilon)\sigma_{pq}^x((l+1)\varepsilon)\sigma_{pq}^x(l\varepsilon) \right.
$$
$$
\left. + \varepsilon h_x \sum_{pq} \eta_p^x(l\varepsilon)\sigma_{pq}^x(l\varepsilon)\eta_q^x(l\varepsilon) + k_2 \sum_p s_p^x(l\varepsilon)\eta_p^x((l+1)\varepsilon)\eta_p^x(l\varepsilon) \right). \tag{22}
$$

Notice that it has four terms: two equal-time terms coming straight through from the $\hat{H}_x$ part of the original, 2$D$ quantum Hamiltonian, and two terms originating from the evaluation

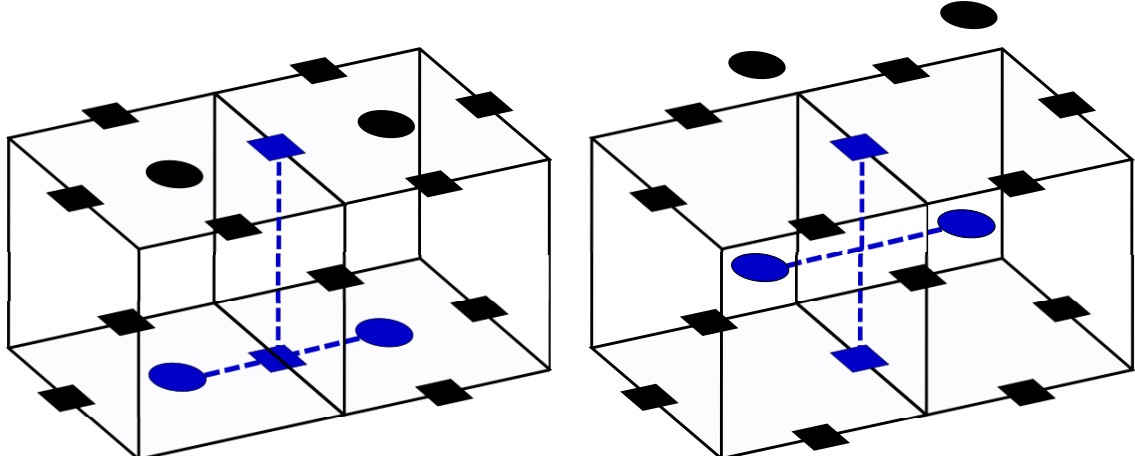

Figure 2: **Renaming $s_p^x$ creates a temporal star.** *Left:* Configurations of $\sigma^x$ (squares) and $s^x$ (circles) classical spins at two time slices $l\varepsilon$ and $(l+1)\varepsilon$ (the horizontal layers of the 3D lattice). Identifying the two plaquettes in the lower layer as $p$ and $q$, the blue spins connected by dashed lines illustrate the $s_p^x(l\varepsilon)s_q^x(l\varepsilon)\sigma_{pq}^x((l+1)\varepsilon)\sigma_{pq}^x(l\varepsilon)$ bond from the $k_2'$ term of $\beta'H_{3D}$. *Right:* We rename the $s_p^x$ spins to $\sigma_{p,p+1}^x$, shifting them in the lattice so that they sit between each plaquette $p$ at $l\varepsilon$ and $p$ at $(l+1)\varepsilon$. The $k_2'$ term illustrated in blue becomes $\sigma_{p,p+1}^x(l\varepsilon)\sigma_{q,q+1}^x(l\varepsilon)\sigma_{pq}^x((l+1)\varepsilon)\sigma_{pq}^x(l\varepsilon)$ and can be clearly recognized as a temporal "star" term. Notice that all such temporal stars are centered around the plane between two cubes of the lattice, rather than about a vertex like spatial stars.

of $\hat{H}_y$ in the matrix element. These two new terms prescribe interactions between spins at *different times* ($l\varepsilon$) and (($l+1)\varepsilon$). Notice that one of these, the new $k_2$ "equal-space" terms, is structurally very similar to the equal-time $h_x$ term, that is, the external field term. To make the correspondence more explicit, we will rename the $s_p^x$ degrees of freedom $\sigma_{p,p+1}^x(l\varepsilon)$, that is, we declare that they are spins living between a plaquette at one time $l\varepsilon$ and the plaquette at the next time $(l+1)\varepsilon$. This is just a renaming: as depicted in Fig. 2 and Fig. 3 , we are systematically shifting where we imagine the $s_p^x$ spins to be on the lattice, which does not change the physics. Under this renaming, the Hamiltonian becomes

$$\beta'H_c = -\sum_{l=0}^{M-1}\left(\varepsilon J_x\sum_s\prod_{p\in s}\sigma_p^x(l\varepsilon) + k_2'\sum_{pq}\sigma_{p,p+1}^x(l\varepsilon)\sigma_{q,q+1}^x(l\varepsilon)\sigma_{pq}^x((l+1)\varepsilon)\sigma_{pq}^x(l\varepsilon)\right.$$
$$\left.+\varepsilon h_x\sum_{pq}\eta_p^x(l\varepsilon)\sigma_{pq}^x(l\varepsilon)\eta_q^x(l\varepsilon) + k_2\sum_p\sigma_{p,p+1}^x(l\varepsilon)\eta_p^x((l+1)\varepsilon)\eta_p^x(l\varepsilon)\right). \tag{23}$$

Now we can explicitly see that the $k_2$ term is exactly another matter field term, or a "line bond," just oriented in the temporal direction. Similarly, we see that the $k_2'$ adds a new temporal $A_s$ which is distinct from the spatial $A_s$ in that it is not centered about a vertex (Fig. 2). Next, we absorb the sum over the discrete time coordinate $l$ into our sums. That is, if we expand the definitions of our sums over spatial and temporal stars (respectively, line bonds) to include spatial and temporal stars (line bonds) existing at *all* times, we obtain,

$$\beta'H_c = -\varepsilon J_x\sum_{s,spatial}A_{s,spatial} - k_2'\sum_{s,temporal}A_{s,temporal}$$
$$-\varepsilon h_x\sum_{pq}\eta_p^x\sigma_{pq}^x\eta_q^x - k_2\sum_{p,p+1}\eta_p^x\sigma_{p,p+1}^x\eta_{p+1}^x. \tag{24}$$

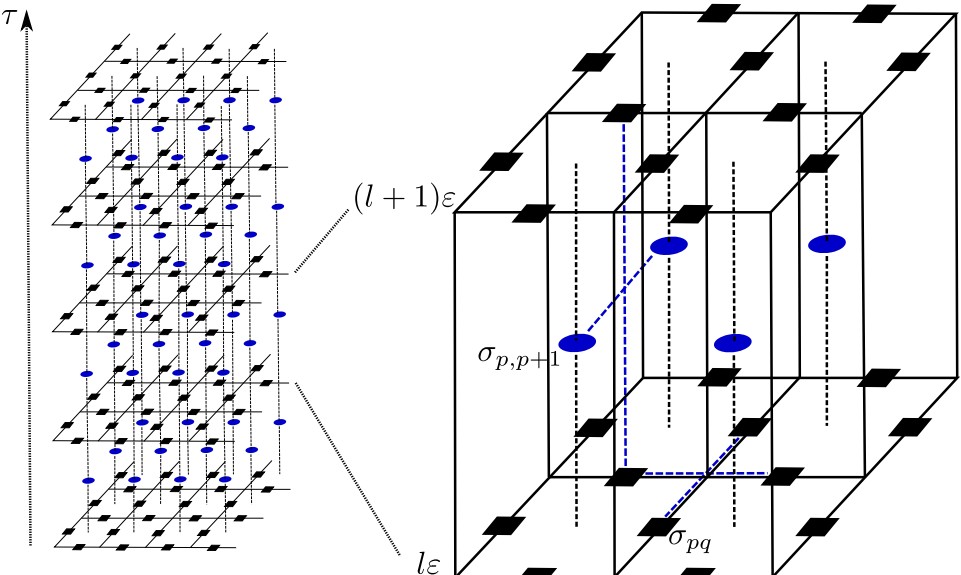

Figure 3: *Left:* After implementing the Suzuki-Trotter decomposition, we obtain a partition function describing classical interacting spins in $3D$. The third dimension corresponds to spin values at point of time $k\Delta\tau$ and $(k+1)\Delta\tau$ in the Suzuki-Trotter decomposition, where $k = 0,1,2...M-1$ (see Eq. 6). *Right:* If we focus on one imaginary time period, we can see the new degrees of freedom $s_p^x$, renamed to $\sigma_{p,p+1}$, that emerged in the computation of the partition function. The new spins participate in the classical 3D Hamiltonian in $A_s$-like terms and bond-like terms in the temporal direction.

This Hamiltonian corresponds to an Ising gauge Hamiltonian in $3D$ to which matter fields are added (see Fradkin Eq. (9.76) [10]) and is dual to the classical Hamiltonian shown in Eq. (5) in Ref. [6]. The duality can be demonstrated by rewriting either Hamiltonian on the dual lattice (see Fig. 1), which will change the $A_s$-like operators of Eq. (24) into $B_p$-like plaquette operators, or, like in Fradkin, with the rotation of the basis from $\sigma^x$ back to $\sigma^z$.

Finally, this Hamiltonian can be simplified further by removing the gauge degrees of freedom by fixing all of the now-classical dummy spins to $\eta_p^x = 1$, and removing the sums over $\{\eta_p^x(k\varepsilon)\}$ in the partition function. Figure 3 illustrates the resulting spatial and temporal degrees of freedom. In this Hamiltonian, the matter field terms have been reduced to external fields, while the star terms are unchanged. We thus have

$$
\begin{aligned}
\beta' H_c = &-\varepsilon J_x \sum_{s,spatial} A_{s,spatial} - k_2' \sum_{s,temporal} A_{s,temporal} \\
&- \varepsilon h_x \sum_{pq} \sigma_{pq}^x - k_2 \sum_{p,p+1} \sigma_{p,p+1}^x .
\end{aligned}
\tag{25}
$$

In this form, the Hamiltonian is a simple Ising gauge theory in $3D$ with classical degrees of freedom $\sigma^x$.

## 4 Conclusion

In these notes, we have provided a detailed derivation of the classical 3D Ising gauge theory starting from the 2D quantum toric code Hamiltonian with external fields. As a consequence of this mapping, the 3D gauge theory is expected to capture the universal physics of a variety

of $2D$ quantum systems that exhibit $\mathbb{Z}_2$ spin liquid phases, including some of the recent experimental results on a programmable quantum computer [11]. Written in the classical spin language, the $3D$ gauge theory provides the paradigmatic examples of stable *confined* and *deconfined* phases, and various possibilities for the transitions between them, including the Higgs and confinement transitions. It is also the starting point for numerical studies using classical Monte Carlo simulations, which have been used recently to probe a number of interesting outstanding questions regarding the physics of the model related to topological order, criticality and error correction [5–8]. We hope that the detailed derivation provided here will help facilitate further understanding of this class of models, the phenomena contained within them, and the compelling equivalence of the quantum and classical systems discussed in these notes.

## Acknowledgements

We thank P. Fendley and D. Sehayek for useful discussions.

**Funding information**   This work was supported by the Natural Sciences and Engineering Research Council of Canada (NSERC), the Canada Research Chair (CRC) program, and the Perimeter Institute for Theoretical Physics. Research at Perimeter Institute is supported in part by the Government of Canada through the Department of Innovation, Science and Economic Development Canada and by the Province of Ontario through the Ministry of Colleges and Universities.

## A   Spin identity No. 1

Given a Pauli spin operator $\hat{\sigma}^x$ and integer $n$, the first spin identity is

$$e^{i\frac{n\pi}{2}(1-\hat{\sigma})} = \begin{cases} \mathbb{1} & \text{when } n \text{ is even}, \\ \hat{\sigma}^x & \text{when } n \text{ is odd}. \end{cases} \tag{26}$$

The proof of spin identity No. 1 can be obtained with Taylor expansion.

$$\begin{aligned} e^{i\frac{n\pi}{2}(1-\hat{\sigma}^x)} &= e^{i\frac{n\pi}{2}} e^{-i\frac{n\pi}{2}\hat{\sigma}^x} \\ &= e^{i\frac{n\pi}{2}} \left( \mathbb{1} - i\frac{n\pi}{2}\hat{\sigma}^x + \frac{(-i\frac{n\pi}{2})^2}{2!}\mathbb{1} + ... \right) \\ &= e^{i\frac{n\pi}{2}} \left( \cos(\frac{-n\pi}{2})\mathbb{1} + i\sin(\frac{-n\pi}{2})\hat{\sigma}^x \right) \\ &= \cos^2(\frac{n\pi}{2})\mathbb{1} - \sin(\frac{n\pi}{2})\sin(\frac{-n\pi}{2})\hat{\sigma}^x \\ &= \begin{cases} \mathbb{1} & \text{when } n \text{ is even}, \\ \hat{\sigma}^x & \text{when } n \text{ is odd}. \end{cases} \end{aligned}$$

where in the fourth equality we have expanded $e^{i\frac{n\pi}{2}}$ into sin and cos and used the fact that cross terms won't survive because $n$ is an integer.

## B   Spin identity No. 2

$$\delta(\mathbb{1}-\hat{\eta}_p^z\hat{B}_p) = \frac{1}{2}\sum_{s_p^x=\pm1} e^{i\pi\frac{1-s_p^x}{2}\left[\frac{1-\hat{\eta}_p^z}{2}+\sum_{pq\in p}\frac{1-\hat{\sigma}_{pq}^z}{2}\right]}. \tag{27}$$

To prove this identity, we begin by expanding out the sum over $s_p$ on the RHS, and pulling down the sum in the exponent, giving

$$\frac{1}{2}\sum_{s_p=\pm1} e^{i\pi\frac{1-s_p}{2}\left[\frac{1-\hat{\eta}_p^z}{2}+\sum_{b\in p}\frac{1-\hat{\sigma}_b^z}{2}\right]} = \frac{1}{2}\left(\mathbb{1}+e^{i\pi\left[\frac{1-\hat{\eta}_p^z}{2}+\sum_{b\in p}\frac{1-\hat{\sigma}_b^z}{2}\right]}\right)$$

$$= \frac{1}{2}\left(1+e^{i\pi\frac{1-\hat{\eta}_p^z}{2}}\left[\prod_{b\in p}e^{i\pi\frac{1-\hat{\sigma}_b^z}{2}}\right]\right).$$

This leaves us with only factors of the form $e^{i\frac{\pi}{2}(1-\hat{\sigma}_b^z)}$ and $e^{i\frac{\pi}{2}(1-\hat{\eta}_b^z)}$ which we can evaluate using Eq. (26). Substituting this in, we get a much simpler expression for the RHS of the identity,

$$\frac{1}{2}\sum_{s_p=\pm1} e^{i\pi\frac{1-s_p}{2}\left[\frac{1-\hat{\eta}_p^z}{2}+\sum_{b\in p}\frac{1-\hat{\sigma}_b^z}{2}\right]} = \frac{1}{2}\left(\mathbb{1}+\hat{\eta}_p^z\prod_{b\in p}\hat{\sigma}_b^z\right)$$

$$= \frac{1}{2}\left(\mathbb{1}+\hat{\eta}_p^z\hat{B}_p\right).$$

To see that this is equal to $\delta(\mathbb{1}-\hat{\eta}_p^z\hat{B}_p)$, we will evaluate both sides in the $z$-basis. The RHS gives

$$\langle\eta_p^z\otimes\{\sigma_b^z\}_{b\in p}|\frac{1}{2}\left(\mathbb{1}+\hat{\eta}_p^z\hat{B}_p\right)|\eta_p^z\otimes\{\sigma_b^z\}_{b\in p}\rangle = \frac{1}{2}\left(1+\eta_p^z B_p\right)$$

$$= \begin{cases} 1 & \text{when } \eta_p^z = B_p\,, \\ 0 & \text{when } \eta_p^z \neq B_p\,, \end{cases}$$

as both $\eta_p^z, B_p = \pm1$. Similarly, because we can pull the Kroenecker delta out of the matrix element, the LHS gives

$$\langle\eta_p^z\otimes\{\sigma_b^z\}_{b\in p}|\delta(\mathbb{1}-\hat{\eta}_p^z\hat{B}_p)|\eta_p^z\otimes\{\sigma_b^z\}_{b\in p}\rangle = \delta(1-\eta_p^z B_p)$$

$$= \begin{cases} \delta(0)=1 & \text{when } \eta_p^z = B_p\,, \\ \delta(1)=0 & \text{when } \eta_p^z \neq B_p\,, \end{cases}$$

which matches the RHS as expected. Therefore, as the identity holds in the $z$-basis, it must be true in all bases.

## C   Spin Identity No. 3

$$e^{i\frac{\pi}{2}(1-\hat{\sigma}_b^z)\left[\left(1-\sigma_b^x(l\varepsilon)\right)+\left(1-\sigma_b^x((l+1)\varepsilon)\right)\right]} = \mathbb{1}. \tag{28}$$

We construct a proof by noticing that, because the $x$-eigenvalues $\sigma_b^x(l\varepsilon), \sigma_b^x((l+1)\varepsilon) = \pm1$, we can immediately evaluate the numerical factor in the brackets, giving

$$e^{i\frac{\pi}{2}(1-\hat{\sigma}_b^z)\left[\left(1-\sigma_b^x(l\varepsilon)\right)+\left(1-\sigma_b^x((l+1)\varepsilon)\right)\right]} = \begin{cases} \mathbb{1} & \text{when } \sigma_b^x(l\varepsilon)=\sigma_b^x((l+1)\varepsilon)=1\,, \\ e^{i2\pi(1-\hat{\sigma}_b^z)} & \text{when } \sigma_b^x(l\varepsilon)=\sigma_b^x((l+1)\varepsilon)=-1\,, \\ e^{i\pi(1-\hat{\sigma}_b^z)} & \text{when } \sigma_b^x(l\varepsilon)\neq\sigma_b^x((l+1)\varepsilon)\,. \end{cases}$$

By using Eq. (26), we can immediately see that all three cases are the identity.



SciPost Phys. Lect. Notes 57 (2022)

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
