# Peer review of "Quantum to classical mapping of the two-dimensional toric code in an external field"

_SciPost Physics Lecture Notes, doi:SciPost Phys. Lect. Notes 57 (2022)_

## Round 1 · Referee Report · Anonymous (Referee 1) · 2022-1-28

Strengths

  1. The Hamiltonian investigated in the manuscript is of great interest for its possible connection to topological phenomena. The availability of a path-integral mapped version is therefore of interest for future theoretical and computational efforts.

  2. The derivation is thorough and complete, except for a few issues of clarity (see "Weaknesses").

Weaknesses

I find that the authors could give a few more useful details in crucial points of the derivation; a detailed list is provided in "Requested changes".

Report

Due to the interest of the topic, as well as the overall clarity of the presented arguments, I would recommend publication in SciPost Lecture Notes once the (relatively minor) issues mentioned in the " Requested changes" sections have been properly addressed.

Requested changes

1) In page (2), the authors mention that new vertex and plaquette spin degrees of freedom can be introduced. For the sake of clarity, I think it would be better to explicitly define these fields, and specify the form of the Hamiltonian in terms of these new variables.

2) In page (3), the authors mention that they introduce redundant spins and that they perform a unitary mapping which transforms the original Hamiltonian into one where the new degrees of freedom are not redundant, but are now coupled with the site spins. The unitary nature of the transformation ensures the invariance of the partition function. The mapping is then chosen in such a way that $\mathbb{Z}_2$ gauge invariance is ensured.

If I understood correctly, these steps are necessary (or at least convenient) to obtain a gauge-invariant classical equivalent of the original model, but I would find it useful to have some details about why it is so (in particular, what would happen by merely doing the Suzuki-Trotter breakup on the original model, and why this approach would be less suitable).

3) At page (5), the authors separate the contribution from two of the terms of the Hamiltonian from the rest of the exponential. As the procedure is presented as exact, I suppose these terms commute with the rest, ensuring no BCH truncation is necessary. For the sake of clarity, I would suggest commenting on which terms of the Hamiltonian commute with each other after presenting eq. (4).

4) In page (8), the authors introduce two new vertex variables, $A_{s, spatial}$ and $A_{s, temporal}$, referring to Fig. 2 for a graphical explanation of their definition. I would find it useful to have the variables depicted in the figure denoted more clearly, along with the original spin degrees of freedom composing them.

(Very) minor issues:

M1) The authors wrote "anzatz" in page (6).

M2) I would suggest inserting row numbering during the submission process, as it would make the review smoother on both sides.

  • validity: top
  • significance: good
  • originality: good
  • clarity: good
  • formatting: perfect
  • grammar: perfect

Author:  Roger Melko  on 2022-05-09  [id 2448]

(in reply to Report 1 on 2022-01-28)
Category:
answer to question

Response to reviewer questions and comments

We thank the referee for the constructive comments on our paper. We appreciate the questions and suggestions, and we have tried to answer and add additional information for all of them. We sincerely hope the modifications will improve the quality of the paper. The new content is marked in italics in the below answers to the reviewer.

Requested change 1

''In page (2), the authors mention that new vertex and plaquette spin degrees of freedom can be introduced. For the sake of clarity, I think it would be better to explicitly define these fields, and specify the form of the Hamiltonian in terms of these new variables.''

Response to 1)

Please notice that Hamiltonian stays the same – these spin degrees of freedom are introduced to prepare for the upcoming unitary transformation, but they are initially not present in any components of the $H_{TC}$. To clarify, we have added following sentence on page 2, line 50:

We should observe that introduction of the new spin degrees of freedom does not change the form of the Hamiltonian $\hat{H}_{TC}$, as it does not contain any terms coupled with $\hat{\vec{\mu}}_s$ and $\hat{\vec{\eta}}_p$.

Requested change 2)

''In page (3), the authors mention that they introduce redundant spins and that they perform a unitary mapping which transforms the original Hamiltonian into one where the new degrees of freedom are not redundant, but are now coupled with the site spins. The unitary nature of the transformation ensures the invariance of the partition function. The mapping is then chosen in such a way that $\mathbb{Z}_2$ gauge invariance is ensured.

If I understood correctly, these steps are necessary (or at least convenient) to obtain a gauge-invariant classical equivalent of the original model, but I would find it useful to have some details about why it is so (in particular, what would happen by merely doing the Suzuki-Trotter breakup on the original model, and why this approach would be less suitable).''

Response to 2

We provide more details on this in second paragraph, page 3 (new line 61):

The Suzuki-Trotter expansion can be done in multiple different ways. The key to the procedure being analytically tractable is to pick the basis of the expansion in such a way that the Hamiltonian can be written as a sum of diagonal and off-diagonal components, and that the portion of the partition function corresponding to the off-diagonal component of the Hamiltonian can be analytically computed. One example of such analytically tractable off-diagonal Hamiltonian is a linear combination of spin degrees of freedom without any higher order terms. This is exactly what motivates our following procedure in which we will transform the perturbed Hamiltonian: $\hat{H}=-\sum_{s}\hat{A}_{s}$ into a form containing an off-diagonal component that is linear in all terms. Of course, this does not preclude that other analytical methods are possible.

as well as highlighting on page 4, line 112:

, while the portion $\hat{H}x=-h}\sum_{p,q}\hat{\eta{p}^{x}\hat{\sigma}}^{x}\hat{\eta{q}^{x}-J$}\sum_{s}\hat{A}_{s , diagonal in $x$-basis, does have terms dependent on products of spins

and on line 115:

To take full advantage of this simplification, we will perform the Suzuki-Trotter decomposition in the $x$-basis ${\sigma_b^x}\otimes{\eta_p^x}$, so that $\hat{H}_z$ is the off-diagonal part of the Hamiltonian. That way, the term $\hat{H}_x$ is diagonal, and off-diagonal $\hat{H}_z$ does not have terms dependent on products of spins, making it tractable for computation of partition function.

Requested change 3)

''At page (5), the authors separate the contribution from two of the terms of the Hamiltonian from the rest of the exponential. As the procedure is presented as exact, I suppose these terms commute with the rest, ensuring no BCH truncation is necessary. For the sake of clarity, I would suggest commenting on which terms of the Hamiltonian commute with each other after presenting eq. (4).''

Response to 3)

We have expaneded on this comment and provided more details on the line 128:

...where we use $\rvert_{tr}$ to denote the condition that $\bra{{\sigma_b^x}\otimes{\eta_p^x}(M\varepsilon)} = \bra{{\sigma_b^x}\otimes{\eta_p^x}(0)}$, originating from the trace. Since $\hat{H}'=\hat{H}_x+\hat{H}_z$ we can apply the Baker–Campbell–Hausdorff (BCH) formula, which simplifies the term $e^{-\varepsilon \hat{H}'} \simeq e^{-\varepsilon \hat{H}_x}e^{-\varepsilon \hat{H}_z}$, with the leading correction term proportional to $-\frac{1}{2}\varepsilon^2[\hat{H}_x,\hat{H}_z]$. So the rest of the calculation will be correct up to the order of $\varepsilon^2$, which is acceptable for $M\gg1$ limit (see Eq.~5).

Additionally , since $\hat{H}_x$ component of the hamiltonian is diagonal in the chosen basis, we added a following additional sentence on page 5, lines 134:

Since $\hat{H}_x$ is diagonal in the ${\sigma_b^x}\otimes{\eta_p^x}$ basis, its sumation is trivial, and $Z_{diag}$ is factored out in the partition function.

Requested change 4)

''In page (8), the authors introduce two new vertex variables, $A_{s,spatial}$ and $A_{s,temporal}$, referring to Fig. 2 for a graphical explanation of their definition. I would find it useful to have the variables depicted in the figure denoted more clearly, along with the original spin degrees of freedom composing them.''

Response to 4)

The complete picture is presented in both figures 2 and 3, so we have added following clarification on page 8, line 195:

This is just a renaming: as depicted in Fig. 2 and Fig. 3 , we are systematically shifting where we imagine the $s^x_p$ spins to be on the lattice, which does not change the physics.

We have also clarified in the caption of Figure 3:

Left: After implementing the Suzuki-Trotter decomposition, we obtain a partition function describing classical interacting spins in $3D$. The third dimension corresponds to spin values at point of time $k\Delta\tau$ and $(k+1)\Delta\tau$ in the Suzuki-Trotter decomposition, where $k=0,1,2...M-1$ (see Eq.~6). Right: If we focus on one imaginary time period, we can see the new degrees of freedom
*$s^x_p$, renamed to * $\sigma_{p,p+1}$, that emerged in the computation of the partition function. The new spins participate in the classical $3D$ Hamiltonian in $A_s$-like terms and bond-like terms in the temporal direction.

Minor issues

Regarding other issues, we have fixed spelling of ansatz, and also introduced line numbers.

---

## Round 2 · Author Response

We thank the referee for the constructive comments on our paper. We appreciate the questions and suggestions, and we have tried to answer and add additional information for all of them. We sincerely hope the modifications will improve the quality of the paper. The new content is marked in *italics* in the below answers to the reviewer.

---

## Round 2 · List of Changes

Requested change 1

''In page (2), the authors mention that new vertex and plaquette spin degrees of freedom can be introduced. For the sake of clarity, I think it would be better to explicitly define these fields, and specify the form of the Hamiltonian in terms of these new variables.''

Response to 1)

Please notice that Hamiltonian stays the same – these spin degrees of freedom are introduced to prepare for the upcoming unitary transformation, but they are initially not present in any components of the $H_{TC}$. To clarify, we have added following sentence on page 2, line 50:

We should observe that introduction of the new spin degrees of freedom does not change the form of the Hamiltonian $\hat{H}_{TC}$, as it does not contain any terms coupled with $\hat{\vec{\mu}}_s$ and $\hat{\vec{\eta}}_p$.

Requested change 2)

''In page (3), the authors mention that they introduce redundant spins and that they perform a unitary mapping which transforms the original Hamiltonian into one where the new degrees of freedom are not redundant, but are now coupled with the site spins. The unitary nature of the transformation ensures the invariance of the partition function. The mapping is then chosen in such a way that $\mathbb{Z}_2$ gauge invariance is ensured.

If I understood correctly, these steps are necessary (or at least convenient) to obtain a gauge-invariant classical equivalent of the original model, but I would find it useful to have some details about why it is so (in particular, what would happen by merely doing the Suzuki-Trotter breakup on the original model, and why this approach would be less suitable).''

Response to 2

We provide more details on this in second paragraph, page 3 (new line 61):

The Suzuki-Trotter expansion can be done in multiple different ways. The key to the procedure being analytically tractable is to pick the basis of the expansion in such a way that the Hamiltonian can be written as a sum of diagonal and off-diagonal components, and that the portion of the partition function corresponding to the off-diagonal component of the Hamiltonian can be analytically computed. One example of such analytically tractable off-diagonal Hamiltonian is a linear combination of spin degrees of freedom without any higher order terms. This is exactly what motivates our following procedure in which we will transform the perturbed Hamiltonian: $\hat{H}=-\sum_{s}\hat{A}{s} - \sum}\hat{B{p}-h_x \sum}\hat{\sigma{b}^{x}-h$}\sum_{b}\hat{\sigma}_{b}^{z into a form containing an off-diagonal component that is linear in all terms. Of course, this does not preclude that other analytical methods are possible.

as well as highlighting on page 4, line 112:

, while the portion $\hat{H}x=-h}\sum_{p,q}\hat{\eta{p}^{x}\hat{\sigma}}^{x}\hat{\eta{q}^{x}-J$}\sum_{s}\hat{A}_{s , diagonal in $x$-basis, does have terms dependent on products of spins

and on line 115:

To take full advantage of this simplification, we will perform the Suzuki-Trotter decomposition in the $x$-basis ${\sigma_b^x}\otimes{\eta_p^x}$, so that $\hat{H}_z$ is the off-diagonal part of the Hamiltonian. That way, the term $\hat{H}_x$ is diagonal, and off-diagonal $\hat{H}_z$ does not have terms dependent on products of spins, making it tractable for computation of partition function.

Requested change 3)

''At page (5), the authors separate the contribution from two of the terms of the Hamiltonian from the rest of the exponential. As the procedure is presented as exact, I suppose these terms commute with the rest, ensuring no BCH truncation is necessary. For the sake of clarity, I would suggest commenting on which terms of the Hamiltonian commute with each other after presenting eq. (4).''

Response to 3)

We have expaneded on this comment and provided more details on the line 128:

...where we use $\rvert_{tr}$ to denote the condition that $\bra{{\sigma_b^x}\otimes{\eta_p^x}(M\varepsilon)} = \bra{{\sigma_b^x}\otimes{\eta_p^x}(0)}$, originating from the trace. Since $\hat{H}'=\hat{H}_x+\hat{H}_z$ we can apply the Baker–Campbell–Hausdorff (BCH) formula, which simplifies the term $e^{-\varepsilon \hat{H}'} \simeq e^{-\varepsilon \hat{H}_x}e^{-\varepsilon \hat{H}_z}$, with the leading correction term proportional to $-\frac{1}{2}\varepsilon^2[\hat{H}_x,\hat{H}_z]$. So the rest of the calculation will be correct up to the order of $\varepsilon^2$, which is acceptable for $M\gg1$ limit (see Eq.~5).

Additionally , since $\hat{H}_x$ component of the hamiltonian is diagonal in the chosen basis, we added a following additional sentence on page 5, lines 134:

Since $\hat{H}_x$ is diagonal in the ${\sigma_b^x}\otimes{\eta_p^x}$ basis, its sumation is trivial, and $Z_{diag}$ is factored out in the partition function.

Requested change 4)

''In page (8), the authors introduce two new vertex variables, $A_{s,spatial}$ and $A_{s,temporal}$, referring to Fig. 2 for a graphical explanation of their definition. I would find it useful to have the variables depicted in the figure denoted more clearly, along with the original spin degrees of freedom composing them.''

Response to 4)

The complete picture is presented in both figures 2 and 3, so we have added following clarification on page 8, line 195:

This is just a renaming: as depicted in Fig. 2 and Fig. 3 , we are systematically shifting where we imagine the $s^x_p$ spins to be on the lattice, which does not change the physics.

We have also clarified in the caption of Figure 3:

Left: After implementing the Suzuki-Trotter decomposition, we obtain a partition function describing classical interacting spins in $3D$. The third dimension corresponds to spin values at point of time $k\Delta\tau$ and $(k+1)\Delta\tau$ in the Suzuki-Trotter decomposition, where $k=0,1,2...M-1$ (see Eq.~6). Right: If we focus on one imaginary time period, we can see the new degrees of freedom
*$s^x_p$, renamed to * $\sigma_{p,p+1}$, that emerged in the computation of the partition function. The new spins participate in the classical $3D$ Hamiltonian in $A_s$-like terms and bond-like terms in the temporal direction.

Minor issues

Regarding other issues, we have fixed spelling of ansatz, and also introduced line numbers.

---

## Editorial Decision

published